# Gaps in tropical science from unrepresentative distribution of sampling and citation across natural terrestrial environments

Effective environmental policies for the tropics depend on accurate, representative scientific data. However, there is strong evidence from particular disciplines and regions that existing research is patchily distributed. Here, we show that poor representation of sampling and citation in some biomes and across key environmental gradients from all disciplines for the entire tropics may lead to flawed scientific paradigms and inappropriate policy prescriptions. We map sampling locations and citations from 2 738 published studies in natural terrestrial tropical environments across all disciplines to identify gaps in field sampling effort and research attention. Five ecoregions – all in moist broadleaf forests – generate 22% of the total citations but cover only 3% of the tropical land area. By contrast, drier biomes with low tree cover account collectively for 57% of the tropical area but generate only 20% of total citations. Locations that are drier, colder, with greater plant species richness, lower tree cover and facing greater climate change extremes are under-sampled and under-cited. Our results will help to correct these imbalances to improve the scientific basis for environmental policies across the tropics.

The terrestrial tropics are highly populated[1] and encompass a wide range of valuable yet threatened ecosystems[2–4]. Numerous international initiatives have emerged to mitigate these threats. These initiatives are shaped by broad syntheses of regional knowledge across all disciplines[4–6], drawing on fieldwork by numerous researchers. However, field research effort across the tropics is uneven, with certain areas disproportionately represented while other regions remain relatively overlooked[7–15]. Previous studies have tracked research activity within particular disciplines, regions or time frames[7–15], consistently revealing strong geographic and thematic biases in research effort, citations, peer review and publication. Moreover, site-specific findings may be extrapolated far beyond their original contexts[15,16], exacerbating the risk of inappropriate policy applications. Still missing is a comprehensive, cross-disciplinary overview of the spatial distribution of tropical field research sampling and study citation, and a

robust assessment of how well this distribution represents the full spectrum of environmental variation across the tropics.

Here, we identify 4260 articles featuring primary field data within the tropics. Habitat types with a high degree of anthropogenic influence (urban and agricultural) account for 36% and 32% of sampling locations and citations, respectively. The spatial distributions of sampling and citation across the tropics are dominated by this prevalence of research on heavily impacted environments (Supplementary Fig. 1). While heavily human-impacted environments in the tropics are widespread and important for policy, they are subject to distinct drivers than environments where direct human influence is minimal[17,18]. To focus on the spatial distribution and drivers of sampling and citation across relatively natural environments, we remove studies featuring urban and agricultural habitats for subsequent analyses.

e-mail: daniel.metcalfe@umu.se

We map 6370 field measurements from 2738 published articles representing 89,468 citations, across all disciplines in different natural habitats in terrestrial tropical biomes and ecoregions, and relate their spatial distribution to a selection of key environmental conditions across the tropics. We compile an initial list of studies with a minimum of 1 citation from a keyword search for "trop*" in the article title on the Web of Science database[19]. The search is designed to minimize introduction of spatial biases arising from the search process itself, such that any biases in the identified body of literature likely reflect genuine trends in research effort and attention[20,21] (Supplementary Fig. 2). Each article is scanned by a trained human reviewer to extract geographic coordinates of field sampling sites, article citation data and habitat sampled (including aquatic freshwater). Citation data are included as a proxy for the scientific influence exerted by specific studies[22]. In cases where multiple coordinates for sampling locations are identified in a paper, citations per sampling location are calculated as total paper citations divided by the number of locations identified per paper.

Using this geo-referenced database, we first summarize the distribution of sampling and citation among tropical biomes and ecoregions using a widely held definition of the terrestrial tropics based upon vegetation structure (Table 1)[23]. Then, we compare the frequency distribution of sampling and citation under different environmental conditions with the actual frequency distribution with which the same conditions occur in nature across the tropics. We select the following eight conditions because of their recognized importance either as ecosystem drivers or as ecosystem attributes controlling major processes or services[2,3,24–26]. (i) current mean annual temperature (MAT) and (ii) precipitation (MAP), (iii) projected changes in future mean annual temperature (ΔMAT) and (iv) precipitation (ΔMAP) by 2100 compared to recent conditions (1970-2000) under the SSP 245 scenario in the coupled model inter-comparison project[27], (v) MODIS derived leaf area index (LAI)[28] and (vi) soil organic carbon stock in the upper 2 meters (SOC) from the SoilGrids product[29], and (vii) vascular plant species richness[30] and (viii) mammal and bird species richness[31]. Finally, we use statistical multivariate modeling[21] to highlight tropical regions with combinations of environmental conditions which are not adequately sampled according to our database.

## Results
### Uneven global representation of field research across tropical biomes

The spatial distribution of sampling locations and citations across the tropics is highly variable (Fig. 1). The moist broadleaf forest biome covers around 29% of the tropics (Supplementary Fig. 3, Table 1) but accounts for 68% and 73% of all sampling locations and citations, respectively (Fig. 1, Table 1). The top five most cited ecoregions (Fig. 2, Supplementary data 1) - all in moist broadleaf forests with major field stations and/or resident population centers - account for 11% and 22% of total locations and citations respectively, but cover only 3% of the tropical biome area. By contrast, drier biomes (dry broadleaf forest, deserts and xeric shrublands, grasslands, savannas and shrublands) account collectively for 57% of the tropical area (Table 1) but feature only 21% and 20% of sampling locations and citations respectively (Fig. 1, Table 1). Deserts and xeric shrublands stand out as poorly sampled and cited both in absolute terms, and after correcting for biome area (Table 1). Mangroves are generally frequently sampled and cited given their limited area (Table 1), although the Guinean mangroves in west Africa is one of the least cited tropical ecoregions (Fig. 2, Supplementary Data 1). Flooded grasslands and savannas, and coniferous forests are cited much less than expected given how often they are sampled (Table 1, citation:sampling ratio).

### Regional gaps in tropical research

Current sampling efforts capture some tropical habitats and conditions well, while others remain relatively under-sampled (Fig. 2). Specifically, current sampling locations adequately represent environmental conditions from only around 30% of the tropics, corresponding mainly with the moist broadleaf forest biome, particularly in Asia (Fig. 2). Areas with environmental conditions that are poorly represented by the present distribution of sampling correspond mainly with biomes in drier regions with low tree cover, particularly in Africa (Fig. 2).

### Representativeness of sampling and citation across the tropical environmental space

The observed distribution of research sampling locations and citations with varying MAT, MAP, ΔMAT, ΔMAP, LAI, SOC, vascular plant, mammal and bird species richness are different from the expected distribution based upon the tropical land area characterized by these same conditions (Fig. 3, Supplementary Fig. 3). Specifically, relatively cold (<20 °C MAT) or dry (<1000 mm MAP) locations with low LAI ( < 3 $m^2$ $m^{-2}$) and predicted to face more climate extremes (greater future warming, cooling or increased precipitation) are less sampled and cited than expected given their spatial extent (Fig. 3), which corresponds to the following tropical biomes: dry broadleaf forest, coniferous forest, grasslands, savannas and shrublands (Supplementary Fig. 3). These areas tend to occur at relatively high and

## Table 1 | The distribution of sampling locations and citations across natural terrestrial habitats in tropical biomes

| Biome | Area | Sampling locations | | Citation rate | | Citation: sampling ratio |
|---|---|---|---|---|---|---|
| | % of total | % of total | Density $10^5$ km$^{-2}$ | % of total | Density $10^5$ km$^{-2}$ | |
| **Grasslands, Savannas & Shrublands** | 31.1 | 11.2 | 5.2 | 10.7 | 66 | 1.0 |
| **Moist Broadleaf Forests** | 28.5 | 68.2 | 34.9 | 73.4 | 493 | 1.1 |
| **Deserts & Xeric Shrublands** | 20.2 | 1.1 | 0.8 | 1.3 | 12 | 1.2 |
| **Dry Broadleaf Forests** | 5.6 | 9.1 | 23.5 | 8.1 | 275 | 0.9 |
| **Montane Grasslands & Shrublands** | 4.4 | 0.9 | 3.0 | 1.1 | 48 | 1.2 |
| **Flooded Grasslands & Savannas** | 1.2 | 0.7 | 8.2 | 0.4 | 62 | 0.6 |
| **Coniferous Forests** | 1 | 1.9 | 27.8 | 0.6 | 115 | 0.3 |
| **Mangroves** | 0.5 | 2.1 | 63.4 | 1.9 | 752 | 0.9 |
| **Extra-tropical other*** | 7.5 | 4.8 | 9.4 | 2.6 | 67 | 0.5 |

Sampling and citation values are derived from a database of 2738 articles, representing 6370 sampling locations and 89468 citations. Sampling and citation density are calculated as the total number of samples and citations from studies occurring in each biome, respectively, divided by the area of the corresponding biome. Citation:sampling ratio is the ratio of % citations to % sampling locations. *Non-tropical biomes included within the 100 km buffer around the formally defined tropical area[23].

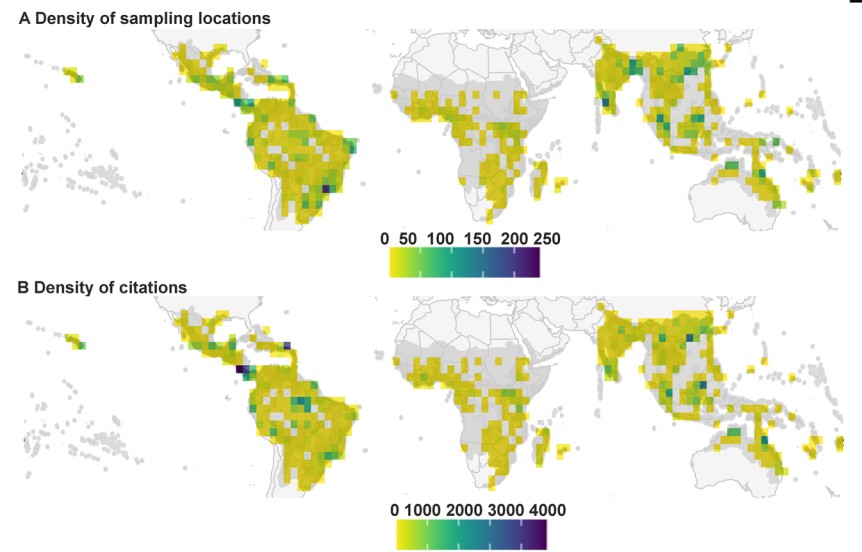

**Fig. 1 | Distribution of field sampling and study citations across natural terrestrial habitats in the tropics.** Density of sampling locations (**A**) and citations (**B**) per unit land area across natural terrestrial habitats in the tropics. Spatial resolution is 3° (~330 km). Maps were produced from a database of 2738 articles, representing 6370 sampling locations and 89,468 citations. The full extent of tropical biomes is highlighted in dark gray, using widely accepted boundaries[23]. To account for transition zones between the biomes, we added a buffer of 100 km around the formally defined tropical area. Overall, the study area consisted of 52.9 × 10⁶ km² of terrestrial land (ca. 36 % of the global land area). Base map from Natural Earth[98].

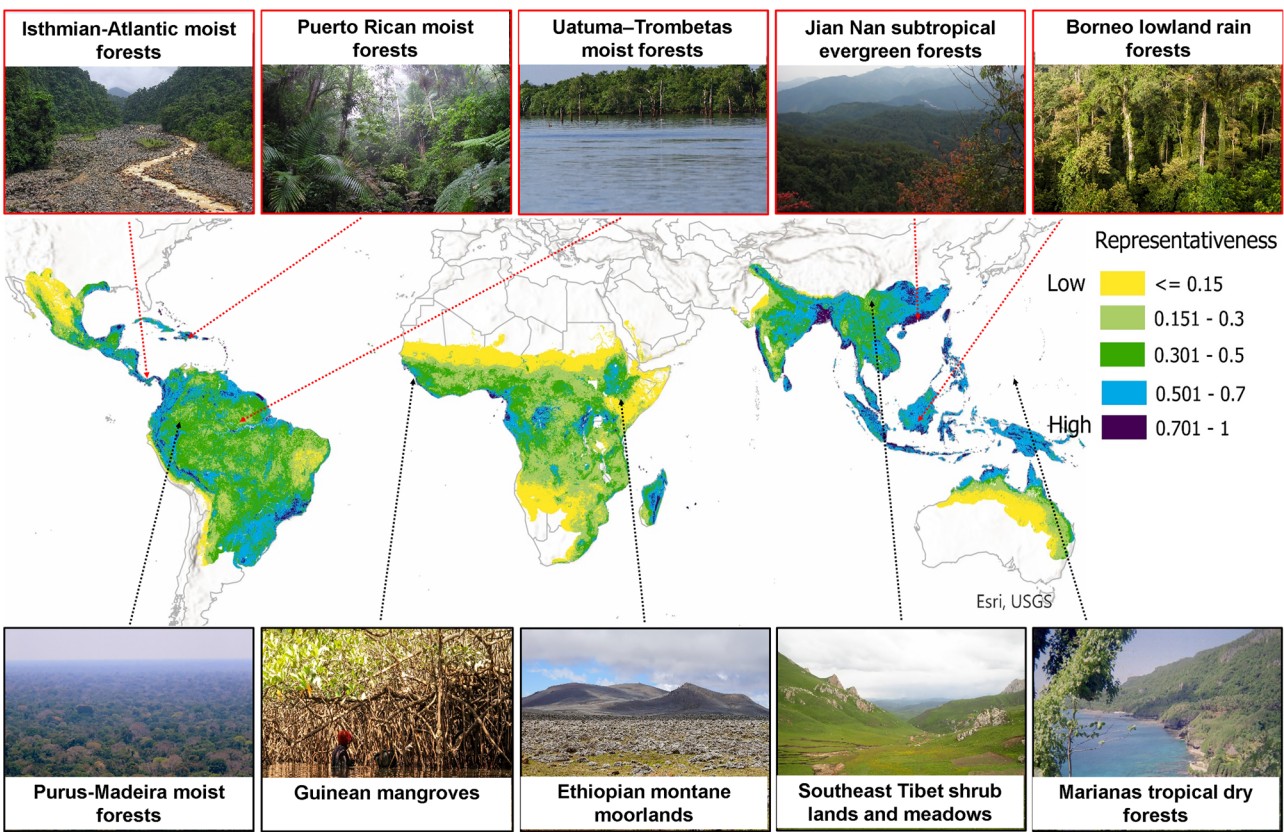

**Fig. 2 | Representativeness of currently sampled environmental conditions across natural terrestrial habitats in the tropics.** Values are derived from a database of 2738 articles, representing 6370 sampling locations and 89,468 citations. Values represent probabilities (1 = high, 0 = low) that environmental conditions within a location have been sampled, using statistical multivariate modeling[16]. A value above 0.5 effectively classifies an environmental condition as one where a sampling location is present. Photographs show the locations of the top five most cited (red outline and arrow) and least cited (black outline and arrow) ecoregions across the terrestrial tropics[23]. Photo credits in Supplementary Table 1. Data of locations and citation metrics across the full list of ecoregions are presented in Supplementary data 1. Base map from Esri[99].

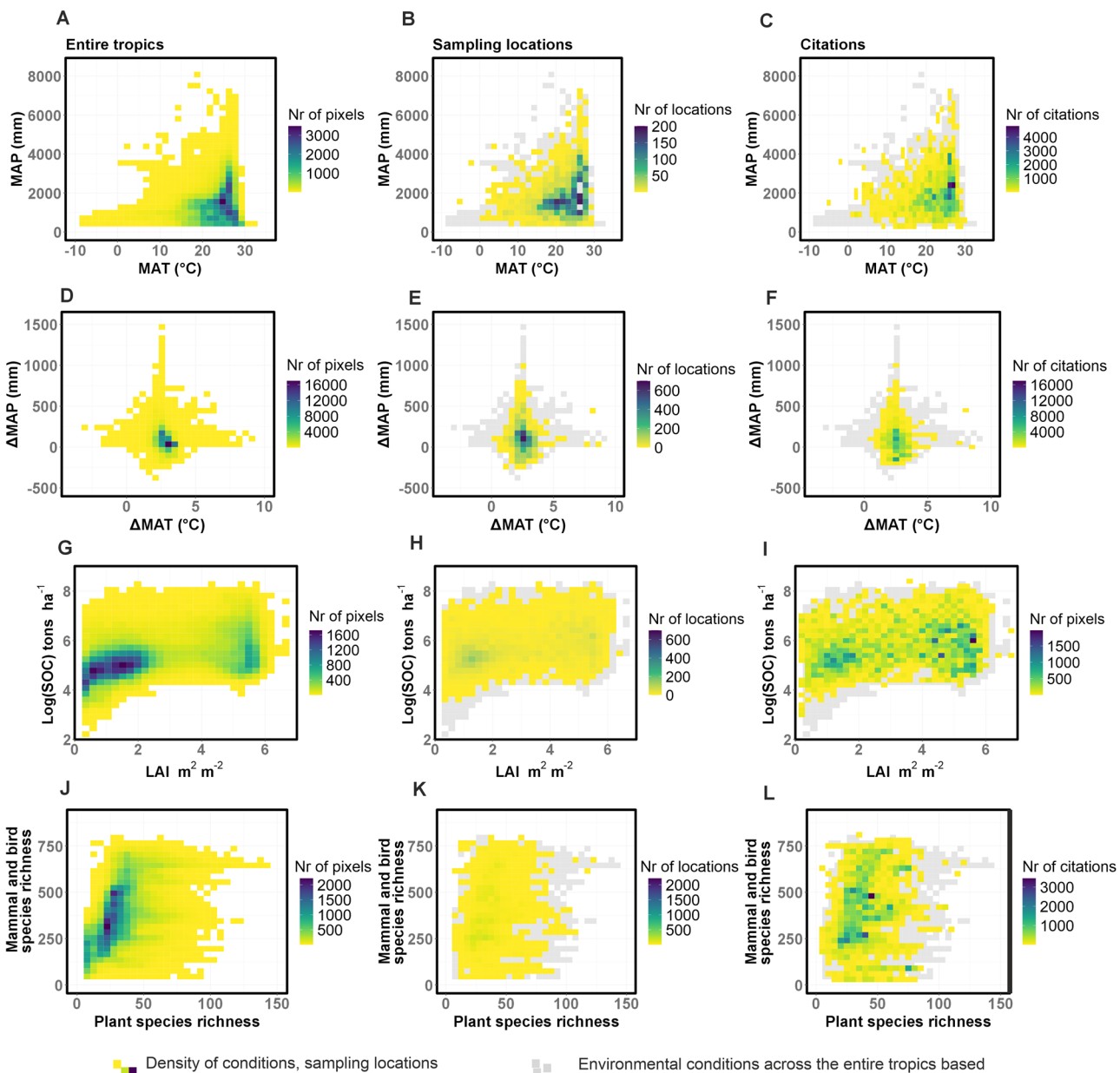

**Fig. 3 | Distribution of field sampling and study citations across environmental space represented by natural terrestrial habitats in the tropics.** Frequency distribution of actual occurrence (**A, D, G, J**), sampling locations (**B, E, H, K**) and citations (**C, F, I, L**) for different combinations of environmental conditions across natural terrestrial habitats in the tropics. Values are derived from a database of 2738 articles, representing 6370 sampling locations and 89,468 citations. The tropics are defined using widely accepted boundaries[23]. Gray pixels denote the full range of ambient conditions across the entire tropics, from a random sample (n = 100 000) of the total pixels within the study area. To be representative, sampling locations (**B, E, H, K**) and citations (**C, F, I, L**) should cover the full range of environmental conditions shown in gray and display a frequency distribution similar to the actual occurrence of environmental conditions (**A, D, G, J**) observed across the entire tropics.

low latitudes, and at higher elevations, within the tropics (Fig. 2). Conversely, areas with high LAI and high diversity of mammals and birds, corresponding roughly with the moist broadleaf forest biome (Supplementary Fig. 3), are sampled and cited more often than expected from the frequency of their occurrence (Fig. 3). It is important to note, however, that the datasets used to quantify actual biodiversity distributions[30,31] are themselves likely spatially biased[32,33], for many of the same reasons that drive sampling and citation biases[7–16]. Therefore, the extent to which actual biodiversity distributions are accurately represented by existing research should be interpreted with caution, though the present analysis likely overestimates representation (Fig. 3).

## Discussion

### Policy risks from unrepresentative sampling and citation in tropical research

Scientific research depends on finite resources, which necessitates difficult decisions about where to focus field sampling efforts. We document major spatial biases in research foci across the terrestrial tropics, which means that valuable ecoregions and widespread environmental conditions remain largely overlooked. For example, the under-sampled and poorly recognized drier biomes show high floristic diversity[34] and play a central role in regulating inter-annual variability in global atmospheric carbon dioxide levels[35]. Yet, these drier biomes are home to around one-third of the global human population[36], their

habitats are generally more threatened[37] and they receive less formal protection than other biomes[38]. By contrast, certain geographic areas, biomes, and ecoregions are disproportionately favored in terms of research effort and attention. These relatively well-sampled and cited regions tend to occur in humid forest biomes, particularly in Asia. The vast diversity of tropical environments exacerbates the risks of extrapolating findings from a narrow set of well-researched contexts to broader, ecologically distinct regions[15,16].

One possible example of such extrapolation is the widespread advocacy – both within and beyond the scientific community – for afforestation in ecosystems with naturally low tree cover as a climate change mitigation strategy[39–41]. The prevalence and persistence of this narrative[16,42,43] may stem, at least in part, from the strong research emphasis we observe in moist broadleaf forest biomes, and the relative scarcity of research in dry forests and open tropical ecosystems such as deserts, grasslands, and shrublands[37,38]. The scientific inferences and policy prescriptions derived from the limited number of intensively sampled locations often stretches far beyond the wider regions which possess clear climatic or ecosystem analogues to the original locations[15,16], hampering the development of effective environmental management actions tailored to suit local conditions.

## Drivers of research imbalances and pathways to more representative insights

As science enters an era of "big data", the urgency to make sense of massive data streams has increased dramatically. One critical challenge is that many large-scale data collection initiatives do not collect representative samples of their variable of interest[44], which means both that the effective sample sizes are much lower and that the mean variable estimates from these samples are inaccurate[45]. The spatial biases we reveal likely emerge from a complex mix of factors: locations of research stations[46], article peer review outcomes and citation rates[8], evolution of population centers and transport infrastructure[1], as well as imbalances among regions in resources available for research[47]. Further, there may be biome-specific differences in the likelihood that research will be referred to as tropical. Although technically a potential methodological artifact in the current analysis, if true, it would nevertheless contribute to the continued marginalisation of certain tropical biomes from policy discussions. As it is, we believe that the trends identified mostly reflect genuine trends in tropical research effort and attention. First, because they are confirmed by multiple independent sources[7–15,37,38]. Second, because the biome most closely linked to the tropics – mangroves[48,49] – where there should be the weakest incentive to specify the tropical origin of the research, is more sampled and cited relative to its extent, not less as would be expected if the search term in our literature review introduced sampling artifacts.

We emphasize that the spatial biases identified are an emergent property resulting from synthesizing many individual research studies, then drawing broad conclusions from them ("external validity" in reviews and meta-analyses[50]), even though individual studies may not make inferences beyond their immediate study site. As such, our results make no claim about the accuracy and quality of individual articles ("internal validity"[50]). Nor do our results suggest that intensively studied research sites and field stations are inherently problematic or not deserving of investment. On the contrary, these infrastructures enable in-depth investigations that would be difficult to execute elsewhere and often yield a high return on investment[51]. Instead, we advocate for complementing the detailed, long-term perspectives provided by intensive research sites with broader pan-tropical perspectives from spatially extensive measurement networks when formulating integrative outputs intended to inform policy. Such networks have already been established to address these challenges, mainly focused on carbon cycling[52,53] and species occurrence[54,55]. While these networks may also be affected by problems associated with unrepresentative spatial sampling[56,57], they remain essential tools for broadening scientific perspectives. More networks addressing other biomes[38], ecosystem components and processes are developing, and will contribute to a more balanced picture of pan-tropical processes as long as the underlying spatial distribution of sampled sites is explicitly considered when deriving broad principles and metrics of tropical ecosystem functioning.

As larger-scale – but often unrepresentative – data collection initiatives flourish, the need to develop strategies to derive accurate, balanced inferences from these datasets is growing ever more urgent. A range of qualitative and quantitative approaches could be used to account for the spatial distribution of sampling[44]. Rigorous assessments of the contributions of bias on descriptive inference - so called "risk-of-bias" assessments – are standard in medical research proposals and papers[58]. Expanding the use of risk-of-bias assessments to other fields could improve scientific transparency and rigor, helping both authors and readers better understand the limitations and generalizability of research findings. Risk-of-bias can be reduced with auxiliary variables which are associated with both the likelihood of a unit being sampled and with the underlying values of the variable of interest, to adjust the overall population-level estimate so that it lies closer to the true value[44,59,60]. Moreover, such variables can also be used to guide future sampling efforts – to target locations which have been sampled less frequently than would be expected by chance[61–63].

The spatial distribution of sampling documented in this study could serve as an auxiliary variable to correct current estimates of ecosystem properties and processes and guide future, more balanced sampling, improving biome or pan-tropical estimates of environmental variables. However, even after statistical and sampling corrections, some residual biases are inevitable. These biases and uncertainties should be clearly communicated to readers and data users to aid interpretation[64,65]. Specifically, the temporal and spatial scope of inference supported by the data should be clearly reported. Where inferences extend beyond the sampled populations or regions, such extrapolations should be explicitly acknowledged and critically assessed. These practices may not currently be incentivized within academia[66] but will become increasingly critical to maintain a clear view of knowns and unknowns in a rapidly changing world inundated with data.

## Toward broader and more inclusive tropical sciences

Our results highlight biomes and environmental conditions that dominate tropical research, and identify priorities for future sampling to improve assessments of the overall current and potential future state of the tropics as a whole. While global disparities and inequities in science and research have received considerable attention[7–16,47,67–69], our study highlights the extent and importance of regional disparities within the tropics[70–72], particularly between South America and Asia versus Africa, and tropical lowland forests versus other habitats. The underlying causes of these regional disparities likely overlap substantially with those driving global patterns: including unequal access to research resources and infrastructure among tropical countries and regions[47,73], variation in social and political stability[74], administrative barriers to knowledge transfer across regions and countries[75], the preferential channeling of international funding and collaborations through a small subset of tropical institutions and countries[46], a bias in research toward forested landscapes relative to other tropical habitat types[70], unethical collaborative practices which disproportionately benefit partners from wealthier regions and/or countries often outside the tropics[76] and systemic biases in the recognition of scientific knowledge production[77,78]. Many of these issues lack straightforward solutions, requiring a paradigm shift in global scientific collaborative practices[76,79]. However, reducing administrative and financial barriers to scientific engagement across different tropical regions and globally – such as the costs of journal subscriptions and the difficulty of securing visas for research visits and study – would represent a major concrete advance[68,75].

Based upon our study, we offer three broad, related but distinct, conclusions and suggestions for future action. First, large portions of the tropics representing valuable ecoregions are relatively well-sampled but poorly cited, and therefore may have had limited influence over scientific narratives or environmental policy. Similar bodies of scientific knowledge originating from different locations receive very different levels of recognition[77]. This issue may be exacerbated by the under-representation of non-English language literature[80,81], which we recommend integrating more fully into future reviews. Fast-evolving translation tools make this a realistic vision[82]. Efforts to increase the diversity of scientific groups – such as journal editors, reviewers, society board members, conference organizers – could help to increase representation from under-recognized tropical regions and countries and reduce systemic bias[69,79]. Some journals have begun experimenting with tools designed to reduce discriminatory practices in academic publishing[83]. If adopted more broadly as standard practice, such tools could contribute to the creation of a more equitable scientific landscape.

Second, some tropical areas remain significantly under-sampled despite their broad extent and ecological value. These areas should be prioritized in future research efforts, as an effective means to increase the amount of novel environmental knowledge per unit research investment. Greater recognition of the scientific value of under-sampled regions by governments, research institutions, funding bodies and journal editors or reviewers would help incentivize researchers to undertake the additional costs often inherent to sampling these areas[16,42,70]. International support for local research infrastructures and field stations could be restructured to begin to counteract the accumulated effects of historical preferences for highly accessible locations near lowland tropical forests.

Finally, we highlight disparities in research attention across a few key axes of environmental variation across the tropics, but there remain many other globally or locally critical drivers of ecosystem processes (e.g., anthropogenic influence, geology, soil type, or plant/animal phylogenetic relatedness), which are likely also not well represented by the current distribution of field research. Further work might highlight new priority areas for future sampling or deserving of greater attention, depending on the process or driver in question. Addressing these data gaps is essential for producing truly integrative, globally relevant ecological insights.

## Methods

### Inclusion & ethics

The authorship team comprises a diverse range of nationalities and career stages with a reasonably balanced gender composition. There is, however, a distinct overrepresentation of North European and American institutions, although several members of the team are nationals of tropical countries but are now employed outside of the tropics. In large part, this reflects the fact that much of the group was initially established to complete a conceptually similar article focused on Arctic systems[20]. For the present analysis, considerable effort was made to widen the authorship group, enlisting assistance and inputs from researchers working in tropical countries, with limited success. Therefore, in the present article, we have taken particular care to evaluate and thoroughly describe the diverse perspectives about the patterns and drivers of regional and global variation in knowledge production.

### Literature review

On 3 November 2021, we searched the Web of Science database for articles with the term "trop*" in their title. Wider keyword searches of the abstract or main text were not performed since they yielded an intractably large number of articles. The approach was not designed to yield a complete list of tropical research, but to provide as close to an unbiased subset of tropical research as possible. As such, more specific search terms were avoided since they could introduce biases if particular names or terms were more likely to be used in particular locations or by particular disciplines. Non-English language articles were not screened out but represent a minority of the Web of Science database[84]. Uncited papers were not included because it was assumed that they have not yet exerted much influence over scientific paradigms or policy strategy[22]. We include all studies irrespective of discipline and all time periods, including social sciences and laboratory studies, as long as the geographic origin of the samples was reported.

The resulting initial list of 11,804 cited papers was then screened to assess their relevance to our objectives (see key steps in the screening process in a PRISMA flow diagram format (Supplementary Fig. 2). Of these papers, 11,713 (99% of initially screened papers) were successfully accessed via university institutional access to the publisher in question or by writing to the corresponding author for a personal copy. After full review, papers were excluded (6625, 56.0% of initially screened papers) if: (1) they featured only measurements in marine environments; tidal estuaries were counted as terrestrial and labeled as river habitats; (2) they did not include primary field measurements because they were broad reviews, modeling analyses, the data presented had already been published elsewhere, or the measurements were from laboratory measurements using samples without a clear provenance; (3) the primary field measurements featured were located outside of the tropics and buffer regions as defined in our study[23]. Studies that were not field-based (for example, remote sensing, geographical information science, and modeling analyses) were, in some cases, included where they included 'groundtruthing' field measurements and/or the spatial extent of the study was relatively limited.

After removing papers that did not fulfill these criteria, 5088 (43% of initially screened papers) papers remained, which were subjected to further analysis. Content analysis was used to: (1) extract geographical coordinates of the field measurements. In cases where coordinates were not explicitly provided, we used place or landform names mentioned in the text to determine the approximate coordinates of the field site(s) on Google Maps; (2) classify the habitats sampled within the paper. The habitats featured were forest, grassland, wetland, desert, rocky area, agricultural, urban, lake, and river. Content analysis inevitably included a degree of subjective judgment on the part of the reviewer. All reviewers were trained at least to a university undergraduate level in environmental sciences and received identical review instructions. Individual papers frequently featured multiple habitats and/or single habitats which represented aspects of multiple habitat categories, in which case a maximum of 3 habitats could be assigned to the same sampling location. The information from the content analysis was then paired with basic paper information derived from Web of Science (authors, journal, title, volume and page numbers, science categories and research areas, citations as of 3 November 2021) to form the central dataset for subsequent analyses.

### Mapping study sampling locations and citations

To further define our study domain for spatial analysis, we used the biome boundaries that were classified as tropical in the ecoregions database (i.e. BIOME_NAME field included a word "tropical")[23]. To acknowledge that there might be transition zones between the biomes, we added a buffer of 100 km around the tropical area. Overall, our domain consisted of $52.9 \times 10^6$ km$^{-2}$ of terrestrial land (ca. 36 % of the global land area). After removing articles that were outside this tropical domain, the number of articles, sampling locations, and citations decreased to 4260, 9987, and 131,030, respectively. Finally, to focus on terrestrial environments with lower intensity of direct human influence, we excluded sampling locations in urban and agricultural areas based on the habitat description in the literature database, which resulted in a dataset of 2738 articles, 6370 sampling locations and 89,468 citations for the final analyses.

## Extraction of environmental conditions variables from study site locations

All the data processing and analyses were conducted in R program version 4.4.1[85]. We used a range of climatic, vegetation, soil and biodiversity data to characterize the tropical region as a whole and to extract data to study site locations. From the biome dataset[23], we utilized the variables ecoregion (ECO_NAME) and biome (BIOME_NAME) to broadly classify articles to key ecological domains. We used the 1-km WorldClim v2 mean annual average air temperature (ºC) and annual cumulative precipitation data (mm) over 1970-2000 and 2081-2100 based on the SSP 2.45 scenario, produced from an ensemble of 12 downscaled CMIP6 layers[27]. Climate anomaly layers were calculated based on the difference between 2081-2100 and 1970-2000 layers. We used the MODIS MOD15A2H dataset, which provides 500-meter resolution data on Leaf Area Index (Lai_500m)[28]. We applied quality filtering to exclude poor-quality pixels (included FparLai_QC bit 0 value 0 data, i.e., good quality) and areas affected by clouds (included FparLai_QC bit 3-4 value 0 data, i.e., no clouds). We then calculated annual means over 2002-2023 and filled gaps in the average MODIS LAI map by applying a moving window analysis (window size: 19) with the focal command in the R package terra[86]. Soil organic carbon stock data for the uppermost 2 meters were extracted from the SoilGrids product[29]. We used a dataset of predicted vascular plant species richness (i.e.: alpha diversity) for a plot size of 1000 m$^2$ including forest and non-forest species (ca. 5 km pixel resolution)[30]. This plot size was chosen as it is commonly used when sampling forests. We further extracted predicted bird and mammal species richness datasets[31] and summed them as one animal diversity measure. All the geospatial layers were re-projected to WGS 1984 at 1 km resolution and masked by the climate datasets using the R package terra[86].

## Spatial analyses

We calculated the total number of articles, sampling locations, and citations across biomes and ecoregions. Then, we examined the distribution of sampling locations and citations across the full range of tropical conditions, to compare with the actual prevalence of the same conditions in reality. To describe the conditions across the entire tropics, we took a random sample (n = 100 000) of the total pixels within our study domain.

We used statistical multivariate modeling to highlight areas lacking sampling locations when considering overall environmental variability[21,87]. This approach is conceptually grounded in species distribution models (SDMs)[88]. SDMs define a geographic space based on environmental variables and identify areas where environmental conditions are suitable for a given species. We adapted this framework to evaluate representativeness of sampling locations, aiming to delineate the spatial distribution of environmental conditions across a geographic envelope that reflects the range of environments captured by the current sampling locations[21,87].

We used a binomial/categorical response variable for the presence-absence data (1 = sampling location exists, 0 = sampling location is missing), and climate (MAT and MAP), soil (SOC), vegetation (LAI), and biodiversity (plant and animal species richness) as explanatory variables. Since our database contains information about sampling locations only, we needed to artificially create locations with the absence of sampling. To do this, we followed an established methodology[89], creating a random sample of terrestrial absence locations with the same number of observations as our presence locations (n = 6447) with the R package sp[90]. Then, we obtained spatial data in these randomly sampled locations based on coordinate colocation. These were then combined with the literature database, which resulted in a data frame of 12,894 locations. The predictors in the final data set did not suffer from high multi-collinearities, as the correlations between the predictor variables were <0.70.

We used common statistical and machine learning models – generalized additive models[91], random forest models[92] and generalized boosted regression trees[93] – to predict both the presence-absence

of sampling locations and the probabilities for the presence. To reduce uncertainties associated with individual models, we calculated the median probability across the three models, which was used to describe the representativeness of sampling locations for each raster pixel across the whole tropics. In the final map, high probabilities indicate a good coverage of current sampling locations in similar conditions (1 = high probability that there is a sampling location in such conditions), and low probabilities suggest lack of locations (0 = no probability for a sampling location). From these probabilities, we also calculated the total area capturing the environmental conditions where sampling sites are covered (>0.5).

The performance of the three models and their ensemble was assessed using cross-validation with 99 permutations from which we calculated the area under the curve (AUC) test statistic[94] with the R package ROCR[95]. In the cross-validation procedure, a random sample of 70% of the data was used to test the model fit, and the remaining 30% were used to assess predictive performance. Test statistics were calculated after each permutation to evaluate the ensemble model. AUC scores varied from 0.76 to 0.9, with the mean AUC being 0.83. An AUC value of 1 represents perfect accuracy and 0.5 indicates that the model is no better than random. All the visualizations from the spatial analyses were created with the R package ggplot2[96] and maps with ESRI ArcGIS Pro version 3.0.3.

## Use of AI

Large language models were used to copy-edit existing text, to check for errors in grammar and syntax, and to suggest alternative sentence formulations.

## Reporting summary

Further information on research design is available in the Nature Portfolio Reporting Summary linked to this article.

## Data availability

All data generated in this study have been deposited in a Zenodo repository file, with DOI 10.5281/zenodo.15423742 (https://zenodo.org/records/15423743)[97]. The data can be downloaded from this link by any user, there are no access restrictions. Additional data are presented together with the article in the file Supplementary data 1, which presents sampling and citation data for the full list of ecoregions included within the study area.

## Code availability

All code generated in this study have been deposited in a Zenodo repository file, with DOI 10.5281/zenodo.15423742 (https://zenodo.org/records/15423743)[97]. The code can be downloaded from this link by any user, there are no access restrictions.

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

## Acknowledgements

We thank Maja Sundqvist, David Wardle and Gesche Blume-Werry for insightful comments on the idea and early versions of the manuscript and data. We acknowledge that certain data included in the manuscript are derived from Clarivate™ (Web of Science™). © Clarivate 2021. All rights reserved. We acknowledge the World Climate Research Programme, which, through its Working Group on Coupled Modeling, coordinated and promoted CMIP6. We thank the climate modeling groups for producing and making available their model output, the Earth System Grid Federation (ESGF) for archiving the data and providing access, and the multiple funding agencies who support CMIP6 and ESGF. Funders: Arctic Six Chairs programme (DBM); European research council consolidator grant ECOHERB 682707 (DBM); Swedish research council for sustainable development 2023-00361 (DBM), 2023-00307 (ABT), 2016-20005 (AL); Swedish Research Council 2021-05265 (GW), 2019-01151 (NC), 2022-04565 (PA); Strategic Research Area "Biodiversity and Ecosystem Services in a Changing Climate" (GW, NC); Strategic Research Area "Modeling the Regional and Global Earth system" (NC); US National Science Foundation 1749252 (HC-Q); Villum Young Investigator grant VIL53048 (JT); Smathers Endowment for Tropical Biology (KJF).

## Author contributions

Conceptualization: D.B.M., A.-M.V.; Methodology: D.B.M., A.-M.V.; Investigation: D.B.M., E.A., H.A., E.P.A., A.E.B., D.C.B., H.B., H.C.-Q., N.C., T.C., C.A.L.D., M.E.D., K.J.F., T.C.W., B.C.H., T.D.G.H., M.J., P.K., A.L., D.L., S.L., G.M., M.M., O.J.L.M., N.S., J.S.P., A.B.-T., J.T., O.K.V., M.W., G.W., W.Z., Y.Y., A.-M.V.; Visualization: D.B.M., A.-M.V.; Project administration: DBM; Writing – original draft: DBM; Writing – review & editing: D.B.M., E.A., H.A., E.P.A., A.E.B., D.C.B., H.B., H.C.-Q., N.C., T.C., C.A.L.D., M.E.D., K.J.F., T.C.W., B.C.H., T.D.G.H., M.J., P.K., A.L., D.L., S.L., G.M., M.M., O.J.L.M., N.S., J.S.P., A.B.-T., J.T., O.K.V., M.W., G.W., W.Z., Y.Y., A.-M.V.

## Funding

## Competing interests

The authors declare no competing interests.

## Additional information

Daniel B. Metcalfe [1] ✉, Emily Anders [2], Hanna Axén [1], E. Petter Axelsson [3], April E. Bermudez [4], David C. Bartholomew [1,5], Nathalie Butt [6,7], Hinsby Cadillo-Quiroz [8], Nitin Chaudhary [9,10,11], Timon Callebaut [1], Cecilia A. L. Dahlsjö [12,13], Mirindi Eric Dusenge [14], Kenneth J. Feeley [15], Thomas Cherico Wanger [16,17], Bernice C. Hwang [1,18], Thirze D. G. Hermans [19], Micael Jonsson [1], Paul Kardol [20,21], Arvid Lindh [21], Daniel Lussetti[21], Shubhangi Lamba [22], Gavyn Mewett [23], Myriam Mujawamariya [24], Olivier Jean Leonce Manzi [22,25,26], Norma Salinas [27], Janet S. Prevéy [28], Aida Bargués-Tobella [21,29], Jing Tang [30], Olivia K. Vought [31], Maria Witteman [22], Göran Wallin [22,32], Wenxin Zhang [33], Yan Yan [34,35] & Anna-Maria Virkkala [36,37,38]

[1]Department of Ecology, Environmental and Geoscience, Umeå University, Umeå, Sweden. [2]Natural Resources and the Environment, University of New Hampshire, Durham, NH, USA. [3]Department of Wildlife, Fish and Environmental Studies, Swedish University of Agricultural Sciences, Umeå, Sweden. [4]Department of Biology, University of New Mexico, Albuquerque, NM, USA. [5]Botanic Gardens Conservation International, Richmond, UK. [6]The Nature Conservancy, South Brisbane, Queensland, Australia. [7]School of the Environment, The University of Queensland; St Lucia, Brisbane, Queensland, Australia. [8]School of Life Sciences, and Biodesign Institute, Arizona State University, Tempe, AZ, USA. [9]Centre for Environmental and Climate Science, Lund University, Lund, Sweden. [10]Department of Physical Geography and Ecosystem Science, Lund University, Lund, Sweden. [11]Stockholm Resilience Centre, Stockholm University, Stockholm, Sweden. [12]School of Geography and the Environment, University of Oxford; South Parks Road, Oxford, UK. [13]Leverhulme Centre for Nature Recovery, University of Oxford; South Parks Road, Oxford, UK. [14]Division of Plant Sciences, Research School of Biology, The Australian National University, Canberra, Australia. [15]Biology Department; University of Miami, Coral Gables, FL, USA. [16]Production Technology & Cropping Systems Group, Agroscope, Nyon, Switzerland. [17]Academy of Global Food Economics and Policy, China Agricultural University; Haidian District, Beijing, China. [18]Department of Ecology, University of Innsbruck, Innsbruck, Austria. [19]Wageningen Social and Economic Research, Wageningen University & Research, Wageningen, The Netherlands. [20]Department of Forest Mycology and Plant Pathology, Swedish University of Agricultural Sciences, Umeå, Sweden. [21]Department of Forest Ecology and Management, Swedish University of Agricultural Sciences, Umeå, Sweden. [22]Department of Biological and Environmental Sciences, University of Gothenburg, Gothenburg, Sweden. [23]Department of Geography, University of Exeter; Amory Building, Rennes Drive, Exeter, UK. [24]Department of Biology, College of Science and Technology, University of Rwanda, Kigali, Rwanda. [25]School of Geography, University of Leeds; University Road, Woodhouse, Leeds, UK. [26]Department of Nature Conservation, Kitabi College, Rwanda Polytechnic, Huye, Rwanda. [27]Department of Science, Chemistry Section, and Institute for Nature Earth and Energy; Pontifical Catholic University of Peru, Lima, Perú. [28]Institute of Arctic and Alpine Research, University of Colorado Boulder, Boulder, CO, USA. [29]AGROTECNIO - CERCA Center, Lleida, Spain. [30]Center for Volatile Interactions, University of Copenhagen, Copenhagen, Denmark. [31]Department of Ecology and Evolutionary Biology, University of Michigan, Ann Arbor, MI, USA. [32]College of Agriculture, Forestry and Food Sciences, University of Rwanda, Musanze, Rwanda. [33]School of Geographical and Earth Sciences, University of Glasgow, Glasgow, UK. [34]College of Environmental & Resource Sciences, Zhejiang University, Hangzhou, China. [35]Key Laboratory of Coastal Environment and Resources of Zhejiang Province, School of Engineering, Westlake University, Hangzhou, Zhejiang, China. [36]Woodwell Climate Research Center, Falmouth, MA, USA. [37]Department of Geosciences and Geography, University of Helsinki, Helsinki, Finland. [38]Finnish Meteorological Institute, Helsinki, Finland. ✉e-mail: daniel.metcalfe@umu.se

