## [Transparent Peer Review file · Nature Communications]

Gaps in tropical science from unrepresentative distribution of sampling and citation across natural terrestrial environments

Corresponding Author: Dr Daniel Metcalfe

Version 0:

Reviewer comments:

Reviewer #1

(Remarks to the Author)

Review of NCOMMS-25-46514-T

I enjoyed reading this manuscript and think could make a nice contribution to the literature. The analysis is satisfyingly simple and, to me, clearly demonstrates that sampling is not representative of the wider environment in the tropics. As I see it, the weakest part of the manuscript is that it is not currently grounded in sampling theory. I have suggested some additional literature and left some technical comments that will hopefully improve the paper in this respect. Perhaps most importantly, the term “sampling bias” is used improperly. Sampling bias (i.e. nonignorable or ‘preferential’ sampling) is defined with respect to a specific study variable (Aubry et al., 2024; Boyd et al., 2025; Schouten et al., 2012). Here there is no one study variable, since many studies are considered, so it’s better to stick to the word “representativeness”. It’s probably also worth mentioning the potential biases induced by considering only those studies published in English (Serrano et al., 2025).

Some specific comments:

71: what does robust mean in this context?

72: could potentially remove the sentence starting “However,...”; since the following one highlights the problem anyway.

77: what is an “unrepresentative scientific paradigm”?

98: what do you mean by biases in peer review?

101: so you’re looking at the locations of the fieldwork and also the number of citations given the location of the fieldwork?

103: sentence reads a bit strangely. Would it be better to say you “map 6370 measurements from 2738 published articles that have been cited 89 468 times...”?

108: this is confusing since you go on to show that these papers are not an unbiased subset.

111: the justification for dropping urban studies isn’t very strong imo. But if you will delineate your study in this way, then it should be clear in the title and abstract.

116: It should be clearer at this point that “patterns of citations” refers to the distribution of citations indexed by original study location (if that’s what you are doing ofc).

127: these maps of plant and vertebrate species richness are presumably very geographically biased too. Some consideration of what this means for your results is needed.

149: it might be nice to plot sample vs. population relative frequencies per environmental category in this section. I.e. relative frequencies of sampled locations per category, possibly of citations per category and the actual frequency distribution of categories across the tropics.

178: what’s a representativeness analysis? If you mean the SDM-type analysis described in the methods, then this is a map of predicted probabilities of sampling rather than representativeness. Some would consider the variability in predicted probabilities across pixels the measure of representativeness (Bethlehem et al., 2008). (The idea here is that under simple random sampling, the usual benchmark for ‘representative’ sampling, every pixel would have an equal probability of being included in the sample. A greater departure from equal inclusion probabilities indicates a greater departure from simple random sampling.)

196: *“vertebrate diversity”

215: what do you mean by “translate into valuable ecoregions”?

Sentence starting on line 218 doesn’t make sense grammatically.

221: how? Stratify on these variables when collecting new data (Pescott et al., 2025)? Also see (Henry et al., 2024; Schouten & Shlomo, 2017).

226: how? Need some substance here about the challenges of extrapolation under unrepresentative sampling.

227: this is a really nice example.

238: this section lacks any reference to statistical corrections, which seem like the obvious approach to mitigating the effects of sampling biases. See e.g. (Boyd et al., 2023; Elliott & Valliant, 2017).

240: geographic biases are driven by spatial biases?

244: it's good that you have acknowledged this.

252: this paragraph is closely related to concepts in meta-analysis. It might be worth referring to internal vs. external validity and risk of bias assessments, which are common in evidence synthesis and increasingly in ecology.

273: again, some discussion of how adaptive sampling might work is needed.

Thanks for the opportunity to review this interesting manuscript,
Rob Boyd

Aubry, P., Francesiaz, C., & Guillemain, M. (2024). On the impact of preferential sampling on ecological status and trend assessment. *Ecological Modelling*, 492, 110707. <https://doi.org/10.1016/j.ecolmodel.2024.110707>

Bethlehem, J., Cobben, F., & Schouten, B. (2008). Indicators for the Representativeness of Survey Response. *Statistics Canada's International Symposium Series: Proceedings*, 11.

Boyd, R. J., Botham, M., Dennis, E., Fox, R., Harrower, C., Middlebrook, I., Roy, D. B., & Pescott, O. L. (2025). Using causal diagrams and superpopulation models to correct geographic biases in biodiversity monitoring data. *Methods in Ecology and Evolution*. <https://doi.org/10.1111/2041-210X.14492>

Boyd, R. J., Stewart, G. B., & Pescott, O. L. (2023). Descriptive inference using large, unrepresentative nonprobability samples: An introduction for ecologists. *Ecology*. <https://doi.org/10.1002/ecy.4214>

Elliott, M. R., & Valliant, R. (2017). Inference for nonprobability samples. *Statistical Science*, 32(2), 249–264. <https://doi.org/10.1214/16-STS598>

Henrys, P. A., Mondain-Monval, T. O., & Jarvis, S. G. (2024). Adaptive sampling in ecology: Key challenges and future opportunities. In *Methods in Ecology and Evolution*. British Ecological Society. <https://doi.org/10.1111/2041-210X.14393>

Pescott, O. L., Powney, G. D., & Boyd, R. J. (2025). Adaptive sampling for ecological monitoring using biased data: a stratum-based approach. *Oikos*. <https://doi.org/10.1002/oik.11115>

Schouten, B., Bethlehem, J., Beullens, K., Kleven, Ø., Loosveldt, G., Luiten, A., Rutar, K., Shlomo, N., & Skinner, C. (2012). Evaluating, Comparing, Monitoring, and Improving Representativeness of Survey Response Through R-Indicators and Partial R-Indicators. *International Statistical Review*, 80(3), 382–399. <https://doi.org/10.1111/j.1751-5823.2012.00189.x>

Schouten, B., & Shlomo, N. (2017). Selecting Adaptive Survey Design Strata with Partial R-indicators. *International Statistical Review*, 85(1), 143–163. <https://doi.org/10.1111/insr.12159>

Serrano, F. C., Marconi, V., Deinet, S., Puleston, H., Wiederhecker, H. C., Diaz-Ricaurte, J. C., Farhat, C., Luría-Manzano, R., Martins, M., de Souza, E., Marques-Souza, S., dos Santos Vieira-Alencar, J. P., Valdujo, P., Freeman, R., & McRae, L. (2025). Knowledge from non-English-language studies broadens contributions to conservation policy and helps to tackle bias in biodiversity data. *Journal of Applied Ecology*. <https://doi.org/10.1111/1365-2664.70092>

(Remarks on code availability)
NA

Reviewer #2

(Remarks to the Author)
This is an interesting and timely contribution that provides valuable perspectives on the biases of sampling and biodiversity research across the globe, particularly the undersampling of the tropics where biodiversity is highest and threats are most severe. The paper applies robust methods and approaches to examine these biases. However, there are several key areas that I think require greater emphasis and refinement.

First, the reliance on published literature as the basis for “observations” raises questions. Did the authors cross-reference their findings with GBIF occurrence records or other biodiversity databases that could serve as useful proxies for sampling effort? Such a comparison might strengthen the analysis. This is also an important area that needs some coverage, and how tropical research contributes to this (see suggested reference for tropical Southeast Asia below).

Second, in the section ‘Towards broader and more inclusive tropical sciences’, while the recommendations are useful, they do not fully reflect the realities and challenges faced by tropical biologists. The point about poor citation practices limiting the visibility and influence of tropical scholarship is well taken, but the discussion that follows is underdeveloped and supported by overly generic recommendations. This is a crucial aspect of the study that deserves more detailed analysis, ideally supported by the data presented. I suggest incorporating socioecological and economic factors that shape these patterns and providing concrete suggestions for how such barriers might be addressed in future research.

Additionally, the issue of “helicopter science” is only briefly mentioned but warrants deeper treatment given its centrality to inequities in tropical research. Relatedly, the institutional representation in the study itself appears skewed; only 5 out of 34 institutions involved are based in the tropics or subtropics. This raises concerns about whether the perspectives of tropical scientists are adequately incorporated into both the analysis and interpretation.

Overall, while there are already several works on the inequalities of sampling and effort in the tropics (which were poorly cited here), the manuscript presents a valuable and much-needed global analysis. However, its contribution would be significantly strengthened by a more nuanced and critical discussion of the structural and systemic challenges facing tropical

science, backed by both the authors' analysis and relevant literature. Please also consider the following references, which may help situate the discussion more effectively:

Hughes, A. C., Than, K. Z., Tanalgo, K. C., Agung, A. P., Alexander, T., Kane, Y., Bhadra, S., Chornelia, A., Sritongchuay, T., Simla, P., Chen, Y., Chen, X., Uddin, N., Khatri, P., & Karlsson, C. (2023). Who is publishing in ecology and evolution? The underrepresentation of women and the Global South. *Frontiers in Environmental Science*, 11, 1211211. <https://doi.org/10.3389/fenvs.2023.1211211>

Amano T, Ramírez-Castañeda V, Berdejo-Espinola V, Borokini I, Chowdhury S, Golivets M, et al. (2023) The manifold costs of being a non-native English speaker in science. *PLoS Biol* 21(7): e3002184. <https://doi.org/10.1371/journal.pbio.3002184>

Jarić, I., Diagne, C. and Chowdhury, S. (2025), Moving beyond continents for global and inclusive science. *Front Ecol Environ*, 23: e2851. <https://doi.org/10.1002/fee.2851>

Tanalgo, K. C. (2025). Open and FAIR data sharing are building blocks to bolster biodiversity conservation in Southeast Asia. *Biological Conservation*, 307, 111192. <https://doi.org/10.1016/j.biocon.2025.111192>

(Remarks on code availability)

Version 1:

Reviewer comments:

Reviewer #1

(Remarks to the Author)

Thank you to the authors for addressing my comments. In my view, the paper is now publishable and will make a nice contribution to the literature.

(Remarks on code availability)

Reviewer #2

(Remarks to the Author)

I thank the authors for their effort in revising the manuscript and addressing the key comments raised in the previous round. The paper has now substantially improved, presenting a more balanced integration of information from various geographical contexts. The limitations of the analysis are also now presented alongside the methods and overall approach, making the study easier to follow and reproduce.

However, the manuscript would still benefit from further refinement to improve the coherence and flow of the narrative. A careful polishing of the text and smoother transitions between ideas will enhance readability and overall presentation, making it suitable for publication

(Remarks on code availability)

N/A

Responses to comments

We thank the two reviewers for their thorough and thoughtful comments. We have addressed these comments (see detailed comments below) and extensively revised the manuscript accordingly. We believe that these changes have resulted in a much stronger and more detailed exposition of the study data. We attach both a final revised version of the manuscript (with major additions highlighted in red), and a track-changed version so that the full detailed changes from the original submission are visible. In the text below, responses are in normal text, and text cited from the revised manuscript are presented within quote marks in italics.

Reviewer 1

Comment) I enjoyed reading this manuscript and think could make a nice contribution to the literature. The analysis is satisfyingly simple and, to me, clearly demonstrates that sampling is not representative of the wider environment in the tropics. As I see it, the weakest part of the manuscript is that it is not currently grounded in sampling theory. I have suggested some additional literature and left some technical comments that will hopefully improve the paper in this respect.

Response) Thank you for your positive evaluation and constructive inputs. We have addressed each of your points in detail, see responses below and line numbers denoting changes in the revised manuscript. We believe that these changes have greatly strengthened the manuscript, adding important detail and context.

Comment) Perhaps most importantly, the term “sampling bias” is used improperly. Sampling bias (i.e. nonignorable or ‘preferential’ sampling) is defined with respect to a specific study variable (Aubry et al., 2024; Boyd et al., 2025; Schouten et al., 2012). Here there is no one study variable, since many studies are considered, so it’s better to stick to the word “representativeness”.

Response) We agree that our use of the term sampling bias risks confusing readers, so we have followed your advice and substituted this term with variants of “representativeness”. Depending on the context we have in certain cases chosen “unrepresentative” or “poorly represented”.

Comment) It’s probably also worth mentioning the potential biases induced by considering only those studies published in English (Serrano et al., 2025).

Response) Some non-english studies were included but they were a small minority of the articles retrieved from the bibliographic search. We already briefly mentioned the likely biases introduced by only including English studies (Lines 277-279 in original submission). However, we have now incorporated your suggested reference to add further detail and context (Line 304-307).

“This issue may be exacerbated by the under-representation of non-English language literature^{79,80}, which we recommend integrating more fully into future reviews and syntheses. Fast evolving translation tools make this a realistic vision⁸¹. “

In addition, we have now added the following relevant information when describing the literature search (Lines 341-342):

“Non-english language articles were not screened out but represent a minority of the Web of Science database⁸⁴”

Comment) 71: what does robust mean in this context?

Response) We agree that the use of “robust” in this context was vague and open to interpretation. We have now replaced this word with “accurate” (Line 71) since the term “accuracy” has a more specific and clear meaning in the context of scientific data: <https://ec.europa.eu/eurostat/statistics-explained/index.php?title=Glossary:Accuracy>

Comment) 72: could potentially remove the sentence starting “However,...”; since the following one highlights the problem anyway.

Response) We agree that this sentence was redundant, so we have removed it.

Comment) 77: what is an “unrepresentative scientific paradigm”?

Response) We agree that this phrasing is not clear. We have now replaced it with the following text (Line 75):

“flawed scientific paradigms”

Comment) 98: what do you mean by biases in peer review?

Response) We were referring here to evidence that the peer review process in academic journals may be biased against certain groups. We now include additional descriptions of this phenomenon with references in the following text (Lines 224-227, 286-295):

“The spatial biases we reveal likely emerge from a complex mix of factors; locations of research stations⁴⁶, article peer review outcomes and citation rates², evolution of population centers and transport infrastructure¹⁷, as well as imbalances among regions in resources available for research⁴⁷.”

“The underlying causes of these disparities likely overlap substantially with those driving global patterns: including unequal access to research resources and infrastructure among tropical countries and regions^{47,73}, variation in social and political stability⁷⁴, administrative barriers to knowledge transfer across regions and countries⁷⁵, the preferential channeling of international funding and collaborations through a small subset of tropical institutions and countries⁴⁶, a bias in research toward forested landscapes relative to other tropical habitat

types⁷⁰, unethical collaborative practices which disproportionately benefit partners from wealthier countries often outside the tropics⁷⁶ and systemic biases in the recognition of scientific knowledge production^{77,78}

Comment) 101: so you're looking at the locations of the fieldwork and also the number of citations given the location of the fieldwork?

Response) Yes, that is correct. We believe that citations are useful to include, in addition to sampling locations for the following reasons: (1) because citations are a useful proxy for the scientific influence exerted by specific studies, and the geographic locations at which studies took place, and (2) because patterns of citations may be subject to a range of different biases from those that affect sampling locations. We had described both the rationale for including citation data and how they were calculated for individual locations in the methods section, but we have now moved this text to the introduction of the study to help with understanding the citation data (Lines 127-131):

"Citation data were included as a flawed but valuable proxy for the scientific influence exerted by specific studies²⁷. In cases where multiple coordinates for sampling locations were identified in a paper, citations per sampling location were calculated as total paper citations divided by the number of locations identified per paper."

Comment) 103: sentence reads a bit strangely. Would it be better to say you "map 6370 measurements from 2738 published articles that have been cited 89 468 times..."?

Response) We agree that your formulation is easier to read and understand. The sentence has now been modified to the following (Lines 118-119):

"Here, we map 6 370 field measurements from 2 738 published articles representing 89 468 citations"

Comment) 108: this is confusing since you go on to show that these papers are not an unbiased subset.

Response) We agree that this sounds confusing, so have modified this text to provide more detail and context (Lines 121-125), the references provide descriptions of similar applications of this search methodology:

"We compiled an initial list of studies with a minimum of 1 citation from a keyword searches for "trop" in the article title on the Web of Science database²⁶. The search was designed to minimize introduction of spatial biases arising from the search process itself, such that any biases in the identified body of literature likely reflected genuine trends in research effort and attention^{15,16}"*

Comment) 111: the justification for dropping urban studies isn't very strong imo. But if you will delineate your study in this way, then it should be clear in the title and abstract.

Response) We already specified in the abstract that the study was focused on natural terrestrial environments (Lines 78-79 in original submission), but we have now clarified this also in the title (Lines 1-2), and in all figure/table legends.

Our dataset initially included all environments, including those heavily influenced by anthropogenic activity (classified as either “urban” or “agricultural” habitats). We found that these anthropogenic habitats constituted a large portion of all samples/citations. In one sense, this is an interesting result but it also means that all patterns in Figures 1-3 are dominated by the trends of sampling/citation in anthropogenic habitats, obscuring other important and interesting trends in more natural habitats. We believe that it would be very difficult to dedicate the necessary attention to the separate drivers and trends presented by the different portions of the entire dataset in the context of a single highly space-limited paper. The approach we have taken is to (1) reserve the data from anthropogenic habitats for the full detailed analysis in a separate article which we believe it merits, and (2) present only the data from habitats with minimal anthropogenic influence in the present analysis.

In the revised manuscript, we now (1) present a new figure in the supplementary material, outlining the key patterns shown by the full dataset (including urban and agricultural studies) (Extended Data Fig. S1), (2) add text in the main manuscript (Lines 109-113) presenting an brief overview of the basic patterns shown by the anthropogenically influenced habitats, which we believe helps to explain our decision to exclude these data from the main analyses presented in the manuscript:

“We identified 4 260 cited articles featuring primary field data within the tropics. Habitat types characterized by a high degree of anthropogenic influence (urban and agricultural) accounted for 36% and 32% of sampling locations and citations respectively. Spatial patterns of sampling and citation across the tropics are dominated by this prevalence of research on heavily impacted environments (Extended Data Fig. S1)”

Comment) 116: It should be clearer at this point that “patterns of citations” refers to the distribution of citations indexed by original study location (if that’s what you are doing ofc).

Response) We agree the term 'pattern' was vague. In our revisions, we now use the more specific terms 'distribution' or 'spatial distribution' when referring to our data (e.g., sampling and citation) and retain 'pattern' only for more general discussions in the literature.

Comment) 127: these maps of plant and vertebrate species richness are presumably very geographically biased too. Some consideration of what this means for your results is needed.

Response) We agree. The source for vascular plant species richness (Sabatini et al. 2002) described some attempt at bias correction in the methods section, but we doubt that will have completely removed the likely very strong biases in the underlying biodiversity data. The source for vertebrate species richness (Jenkins et al. 2013) gives no clear description of any correction for unrepresentative sampling. We speculate that the distribution of species data and our measurements of sampling and citation should be highly correlated since they will both likely be confounded by very similar factors affecting research effort. This means that (1) we should be generally cautious about the species maps and our observed relationships between

sampling/citation and diversity, (2) we may overestimate the extent to which current sampling and citation “match” or represent current spatial patterns of diversity, so our current assessment of representativeness of sampling for biodiversity will, if anything, tend to be optimistic. We now include new text outlining these key points in the manuscript, with appropriate references (Lines 188-192):

“It is important to note, however, that the datasets used to quantify actual biodiversity distributions^{13,14} are themselves likely spatially biased^{32,33}, for many of the same reasons that drive sampling and citation biases^{1-9,23}. Therefore, the extent to which actual biodiversity distributions are accurately represented by existing research should be interpreted with caution, though the present analysis likely overestimates representation (Fig. 3).”

Comment) 149: it might be nice to plot sample vs. population relative frequencies per environmental category in this section. I.e. relative frequencies of sampled locations per category, possibly of citations per category and the actual frequency distribution of categories across the tropics.

Response) We believe that the relative frequency per biome of all of these variables – the total population as well as sampled locations and citations – are presented as percent values in Table 1 (and for individual ecoregions in the Supplementary information). However, we may be misunderstanding your comment. We are happy to make additional changes but may require additional guidance on this point.

Comment) 178: what’s a representativeness analysis? If you mean the SDM-type analysis described in the methods, then this is a map of predicted probabilities of sampling rather than representativeness. Some would consider the variability in predicted probabilities across pixels the measure of representativeness (Bethlehem et al., 2008). (The idea here is that under simple random sampling, the usual benchmark for ‘representative’ sampling, every pixel would have an equal probability of being included in the sample. A greater departure from equal inclusion probabilities indicates a greater departure from simple random sampling.)

Response) We agree that this term was vague and liable to create confusion. The approach we have chosen, which we believe is more in line with the rest of the results text, is to remove this initial part of the sentence and only focus on the results. The analyses underlying these results are instead mentioned in the Figure 2 caption (Lines 739-742), and are presented in detail in the Methods section (Lines 419-456).

Comment) 196: **“vertebrate diversity”*

Response) We agree that “animal” is overly broad. However, this refers to data which covers mammals and birds, not all vertebrates. So we have revised the text to the following (Lines 185-188):

“Conversely, areas with high LAI and high diversity of mammals and birds, corresponding roughly with the moist broadleaf forest biome (Extended Data Fig. S3), were sampled and cited more often than expected from the frequency of their occurrence (Fig. 3)”

Comment) 215: what do you mean by “translate into valuable ecoregions”?

Response) We have modified this sentence to hopefully make it simpler and clearer (Lines 196-198):

“We document major spatial biases in research foci across the terrestrial tropics, which means that valuable ecoregions and widespread environmental conditions remain largely overlooked”

Comment) Sentence starting on line 218 doesn’t make sense grammatically.

Response) We have modified this sentence to make it gramatically correct, and clearer that we were referring to the drier biomes mentioned in the preceding sentence (Lines 200-202):

“Yet, these drier biomes are home to around one-third of the global human population³⁶, their habitats are generally more threatened³⁷ and they receive less formal protection than other biomes³⁸”

Comment) 221: how? Stratify on these variables when collecting new data (Pescott et al., 2025)? Also see (Henry et al., 2024; Schouten & Shlomo, 2017).

226: how? Need some substance here about the challenges of extrapolation under unrepresentative sampling.

Response) We now address the issues raised by these two comments with substantial new text and references in the next section (Lines 220-224, 256-278) which is more explicitly targeted at practical strategies to provide more representative insights. So we have chosen to remove the claim that our study will help to focus resources in the section “Policy risks from unrepresentative sampling and citation in tropical research” (Lines 194-217). The content of this section is now better aligned with the described section subheading, focusing only on the observed patterns of sampling and citation and their likely policy consequences.

Comment) 238: this section lacks any reference to statistical corrections, which seem like the obvious approach to mitigating the effects of sampling biases. See e.g. (Boyd et al., 2023; Elliott & Valliant, 2017).

Response) Thank you for pointing us towards this literature in this and the two previous comments, they are indeed highly relevant to the issues we discuss. We now include a large amount of new text describing various practical approaches, with relevant references, to mitigating the effects of unrepresentative sampling (Lines 220-224, 256-278). Broadly, these include (1) greater use of risk-of-bias assessments to acknowledge and counteract biases where possible (Lines 259-263), (2) greater use of statistical sample adjustments and adaptive sampling strategies (Lines 263-268), (3) more explicit description of biases and limitations in the data collected in scientific studies and what it means for scientific inferences to wider regions/populations/time periods (Lines 272-276).

Comment) 227: this is a really nice example.

Response) Thank you.

Comment) 240: geographic biases are driven by spatial biases?

Response) We agree that this is an obvious connection, we have revised the text to the following (Lines 224-227):

“The spatial biases we reveal likely emerge from a complex mix of factors; locations of research stations⁴⁶, article peer review outcomes and citation rates², evolution of population centers and transport infrastructure¹⁷, as well as imbalances among regions in resources available for research⁴⁷.”

Comment) 244: it’s good that you have acknowledged this.

Response) Thanks, yes we think this is an important possibility to acknowledge, although we believe for the reasons outlined in the text it is unlikely to affect our key results (Lines 228-236).

Comment) 252: this paragraph is closely related to concepts in meta-analysis. It might be worth referring to internal vs. external validity and risk of bias assessments, which are common in evidence synthesis and increasingly in ecology.

Response) We now explicitly include the terms internal and external validity, with relevant references, where this topic is introduced (Lines 237-241).

“We emphasize that the spatial biases identified are an emergent property resulting from synthesizing many individual research studies then drawing broad conclusions from them (“external validity” in reviews and meta-analyses⁵⁰), even though individual studies may not make inferences beyond their immediate study site. As such, our results make no claim about the accuracy and quality of individual articles (“internal validity”⁵⁰).”

In the new text describing strategies to account for unrepresentative sampling in scientific studies, we now describe, with references, the basic principle and purpose of risk-of-bias assessments (Lines 259-263).

“Rigorous assessments of the contributions of bias on descriptive inference - so called “risk-of-bias” assessments – are standard in medical research proposals and papers⁵⁸. Expanding the use of risk-of-bias assessments into other fields could improve scientific transparency and rigor, helping both authors and readers better understand the limitations and generalizability of research findings.”

Comment) 273: again, some discussion of how adaptive sampling might work is needed.

Response) We agree. In the new text describing strategies to account for unrepresentative sampling in scientific studies, we now describe, with references, the basic principle and purpose

of adaptive sampling, although we have chosen to not explicitly use the term “adaptive sampling” since it is a rather specialist term which may create confusion and distract from the core point (Lines 263-268).

“Risk of bias can be reduced with auxiliary variables which are associated with both the likelihood of a unit being sampled and with the underlying values of the variable of interest, to adjust the overall population-level estimate so that it lies closer to the true value^{44,59,60}. Moreover, such variables can also be used to guide future sampling efforts – to target locations which have been sampled less frequently than would be expected by chance⁶¹⁻⁶³. The spatial distribution of sampling documented in this study could serve as an auxiliary variable to correct current estimates and guide future, more balanced sampling, improving biome or pan-tropical estimates of environmental variables.”

Reviewer 2

Comment) This is an interesting and timely contribution that provides valuable perspectives on the biases of sampling and biodiversity research across the globe, particularly the undersampling of the tropics where biodiversity is highest and threats are most severe. The paper applies robust methods and approaches to examine these biases. However, there are several key areas that I think require greater emphasis and refinement.

Response) Thank you for your positive evaluation and constructive inputs. We have addressed each of your points in detail, see responses below, resulting in extensive changes in the revised manuscript. We believe that these changes have greatly strengthened the manuscript, adding important detail and context.

Comment) First, the reliance on published literature as the basis for “observations” raises questions. Did the authors cross-reference their findings with GBIF occurrence records or other biodiversity databases that could serve as useful proxies for sampling effort? Such a comparison might strengthen the analysis. This is also an important area that needs some coverage, and how tropical research contributes to this (see suggested reference for tropical Southeast Asia below).

Response) After some careful reflection we have decided not include any formal comparison or analysis using GBIF data in the revised manuscript, for the following reason. There is extensive evidence that GBIF data is spatially biased for many of the same reasons described in our paper (Hughes et al. 2021, Beck et al. 2014, Garcia-Rosello et al. 2023, Bowler et al. 2025). Therefore, we believe that the spatial distribution of sampling and citation would be highly correlated with the GBIF distribution, mainly because they are driven by the same underlying factors. We are not convinced that highlighting this correlation would contribute much useful insight to the analysis. However, we do now explicitly acknowledge the possibility that the biodiversity datasets used in the article may themselves be spatially biased, and include some discussion of how this may affect our interpretations (Lines 188-192):

“It is important to note, however, that the datasets used to quantify actual biodiversity distributions^{13,14} are themselves likely spatially biased^{32,33}, for many of the same reasons that drive sampling and citation biases^{1-9,23}. Therefore, the extent to which actual biodiversity distributions are accurately represented by existing research should be interpreted with caution, though the present analysis likely overestimates representation (Fig. 3).”

Thank you for the reference from tropical Southeast Asia, we have now included it within the text as an excellent example of the strong spatial variation in research effort within different tropical countries and regions (Lines 284-286).

“our study highlights the extent and importance of regional disparities within the tropics^{71,72} – particularly between South America and Asia versus Africa, and tropical lowland forests versus other habitats”

- Hughes, A.C., Orr, M.C., Ma, K., Costello, M.J., Waller, J., Provoost, P., Yang, Q., Zhu, C. and Qiao, H. Sampling biases shape our view of the natural world. *Ecography* **44**, 1259-1269 (2021)
- J. Beck, M. Böller, A. Erhardt, W. Schwanghart. Spatial bias in the GBIF database and its effect on modeling species' geographic distributions. *Ecological Informatics* **19**, 10-15 (2014).
- E. Garcia-Rosello, J. Gonzalez-Dacosta, C Guisande, and J. M. Lobo. GBIF falls short of providing a representative picture of the global distribution of insects. *Systematic Entomology* **48**, 489–497 (2023).
- D. E. Bowler, R. J Boyd, C. T Callaghan, R. A Robinson, N. J. B. Isaac and M. J. O. Pocock. Treating gaps and biases in biodiversity data as a missing data problem. *Biological Reviews* **100**, 50-67 (2025).

Comment) Second, in the section ‘Towards broader and more inclusive tropical sciences’, while the recommendations are useful, they do not fully reflect the realities and challenges faced by tropical biologists. The point about poor citation practices limiting the visibility and influence of tropical scholarship is well taken, but the discussion that follows is underdeveloped and supported by overly generic recommendations. This is a crucial aspect of the study that deserves more detailed analysis, ideally supported by the data presented. I suggest incorporating socioecological and economic factors that shape these patterns and providing concrete suggestions for how such barriers might be addressed in future research.

Response) We thank the reviewer for pushing us to confront these issues more directly, we believe that the manuscript is greatly improved as a result. We agree that the underlying social and economic drivers of the observed patterns are critical to study. However, we believe that we cannot do justice to the complexity and importance of this topic with a formal quantitative analysis in the present manuscript, where it would be presented as supplementary to the main topic, and distract from the core focus on environmental properties surveyed in tropical science. Instead, we plan to address these drivers in a separate, dedicated analysis and manuscript.

Nevertheless, in the present manuscript we have now added substantial new text to this section that provides a much more detailed description of different key issues, including socioecological and economic factors, driving differences in scientific productivity and recognition among different tropical countries and regions (Lines 283-299, 303-304, 307-312, 316-321). Together with this text, we have added, where relevant, multiple new references (references 67-84) which describe in detail the diverse challenges facing tropical scientists, and the broader dynamics driving global and regional inequity in scientific knowledge production.

Specifically, we now provide a much more detailed overview of the drivers of inequities (Lines 283-295). Further, we now specify some possible reforms including lowering of financial and administrative barriers in academic publishing and research (Lines 296-299), improving representation in academia (Lines 307-312), and restructuring global funding and collaborative networks led by wealthier country partners away from the small number of tropical research

institutions and field stations which currently dominate to a wider range of countries and regions (Lines 316-321).

Comment) Additionally, the issue of “helicopter science” is only briefly mentioned but warrants deeper treatment given its centrality to inequities in tropical research. Relatedly, the institutional representation in the study itself appears skewed; only 5 out of 34 institutions involved are based in the tropics or subtropics. This raises concerns about whether the perspectives of tropical scientists are adequately incorporated into both the analysis and interpretation.

Response) We agree, we now refer to a wide range of inequities including “unethical collaborative practices which disproportionately benefit partners from wealthier countries often outside the tropics”, which includes the phenomenon of helicopter science (Lines 292-294):

“The underlying causes of these disparities likely overlap substantially with those driving global patterns: including unequal access to research resources and infrastructure among tropical countries and regions^{47,73}, variation in social and political stability⁷⁴, administrative barriers to knowledge transfer across regions and countries⁷⁵, the preferential channeling of international funding and collaborations through a small subset of tropical institutions and countries⁴⁶, a bias in research toward forested landscapes relative to other tropical habitat types⁷⁰, unethical collaborative practices which disproportionately benefit partners from wealthier countries often outside the tropics⁷⁶ and systemic biases in the recognition of scientific knowledge production^{77,78}.”

We acknowledge that our author composition reflects many of the problems highlighted. These problems are, as we say in the article, relatively easy to identify but in many cases difficult to solve (Lines 295-296). In our case, the northern bias of the author list largely derives from the fact that much of the same author team were involved several years ago in a conceptually very similar paper focused on northern systems (Metcalf et al. 2018. *Nat Ecol Evol* 2, 1443–1448). We note also that multiple authors with affiliations at non-tropical institutions originate from tropical regions, so have a strong personal connection with tropical science despite their current affiliation. Given our author group’s northern bias, we have taken great effort to examine the wide range of published perspectives on this issue before adding our own contribution. We believe that the current more detailed and extensively referenced discussion text is a fair overview of the key issues and potential solutions that have already been presented by tropical scientists. Furthermore, we now discuss the background of the group and the ethics raised in a new Inclusion & Ethics section (Lines 710-721)

*“The authorship team comprises a diverse range of nationalities and career stages with a reasonably balanced gender composition. There is, however, a distinct overrepresentation of north European and American institutions, although, several members of the team are nationals of tropical countries but are now employed outside of the tropics. In large part, this reflects the fact that much of the group was initially established to complete a conceptually similar article (Metcalf et al. 2016. *Nature Ecology & Evolution* 2: 1443-1448) focused on Arctic systems. For the present analysis, considerable effort was made to widen the authorship group enlisting assistance and inputs from researchers working in tropical countries, with limited success. Therefore, in the present article we have taken particular care to evaluate and thoroughly describe the diverse perspectives about the patterns and drivers of regional and global variation in knowledge production.”*

Comment) Overall, while there are already several works on the inequalities of sampling and effort in the tropics (which were poorly cited here), the manuscript presents a valuable and much-needed global analysis. However, its contribution would be significantly strengthened by a more nuanced and critical discussion of the structural and systemic challenges facing tropical science, backed by both the authors' analysis and relevant literature. Please also consider the following references, which may help situate the discussion more effectively:

Response) We have now added substantial new text to this section that provides a much more detailed description of different key issues, including socioecological and economic factors, driving differences in scientific productivity and recognition among different tropical countries and regions (Lines 283-299, 303-304, 307-312, 316-321). Together with this text, we have added, where relevant, multiple new references including those you provide (references 67-84) which describe in detail the diverse challenges facing tropical scientists, and the broader dynamics driving global and regional inequity in scientific knowledge production.

Specifically, we now provide a much more detailed overview of the drivers of inequities (Lines 283-295). Further, we now specify some possible reforms including lowering of financial and administrative barriers in academic publishing and research (Lines 296-299), improving representation in academia (Lines 307-312), and restructuring global funding and collaborative networks led by wealthier country partners away from the small number of tropical research institutions and field stations which currently dominate to a wider range of countries and regions (Lines 316-321).